# A Possible Pathogenic PSEN2 Gly56Ser Mutation in a Korean Patient with Early-Onset Alzheimer’s Disease

**DOI:** 10.3390/ijms23062967

**Published:** 2022-03-09

**Authors:** Kyu-Hwan Shim, Min-Ju Kang, Heewon Bae, Danyeong Kim, Jiwon Park, Seong-Soo A. An, Da-Eun Jeong

**Affiliations:** 1Department of Bionano Technology, Gachon University, Seongnam-si 13120, Korea; smuller0305@gmail.com (K.-H.S.); dan627328@gmail.com (D.K.); 2Department of Neurology, Veterans Health Service Medical Center, Veterans Medical Research Institute, Seoul 05368, Korea; minju0321@naver.com (M.-J.K.); hwbae0601@gmail.com (H.B.); 3Department of Psychology, University of Utah, Salt Lake City, UT 84112, USA; andrewjp0219@gmail.com

**Keywords:** Alzheimer’s disease, mutation, presenilin 2, phosphorylation, hippocampal atrophy, PSEN2 Gly56Ser mutation

## Abstract

Early-onset Alzheimer’s disease (EOAD) is characterized by the presence of neurological symptoms in patients with Alzheimer’s disease (AD) before 65 years of age. Mutations in pathological genes, including *amyloid protein precursor* (*APP*), *presenilin-1* (*PSEN1*), and *presenilin-2* (*PSEN2*), were associated with EOAD. Seventy-six mutations in *PSEN2* have been found around the world, which could affect the activity of γ-secretase in amyloid beta processing. Here, a heterozygous *PSEN2* point mutation from G to A nucleotide change at position 166 (codon 56; c.166G>A, Gly56Ser) was identified in a 64-year-old Korean female with AD with progressive cognitive memory impairment for the 4 years prior to the hospital visit. Hippocampal atrophy was observed from magnetic resonance imaging-based neuroimaging analyses. Temporal and parietal cortex hypometabolisms were identified using fluorodeoxyglucose positron emission tomography. This mutation was at the N-terminal portion of the presenilin 2 protein on the cytosolic side. Therefore, the serine substitution may have promoted AD pathogenesis by perturbing to the mutation region through altered phosphorylation of presenilin. In silico analysis revealed that the mutation altered protein bulkiness with increased hydrophilicity and reduced flexibility of the mutated region of the protein. Structural changes were likely caused by intramolecular interactions between serine and other residues, which may have affected APP processing. The functional study will clarify the pathogenicity of the mutation in the future.

## 1. Introduction

Alzheimer’s disease (AD) is the most common type of neurodegenerative disorders. AD is characterized by the presence of senile plaques and neurofibrillary tangles, which are composed of aggregated amyloid beta (Aβ) and tau proteins in the brain [1]. Aβ peptides, the most important causative factor of AD, induces neurotoxicity by forming oligomers via self-aggregation [2]. Aβ may also trigger microglial activation and reactive astrocyte proliferation, leading to a release of proinflammatory cytokines, such as interleukin (IL) 1β and tumor necrosis factor (TNF)-α, inducing neuroinflammation [3]. In addition, Aβ could cause neuronal cell death and cognitive impairment through several mechanisms, including glutaminase downregulation and Ca^2+^ flux induction [4]. A recent study addressed that the structural and physiological changes in the frontal and medial temporal lobes under the effect of normal aging determined cognitive alternation [5], which could be exacerbated by intrinsic or extrinsic properties of Aβ. Previous studies demonstrated that changes in the Aβ peptide and γ-secretase complex could alter the aggregation properties and Aβ production, resulting in the increased amount of Aβ oligomer formations. Therefore, mutations in the amyloid precursor protein (*APP*) and components of the γ-secretase complex may facilitate AD by increasing the levels of Aβ peptide aggregation and production. Aβ would be deposited mainly in the following brain regions: the hippocampus, the amygdala, and the entorhinal cortex [6]. Although the cognitive impairment in AD was characterized by transient memory impairment due to temporal lobe damage, recent studies have suggested the association between different brain regions [7]. Abnormal cognitive processes in AD patients could be correlated with Aβ accumulations in the prefrontal cortex (PFC) and other relevant brain regions [8]. The anterior cingulate cortex (ACC) and the PFC were consistently associated with salience detection, error prediction, error monitoring, monitoring of undesirable outcomes, and major depressive disorders [9,10,11,12]. The ACC and the PFC were the key components of cognitive networks and were activated routinely during the monitoring of the external environments and maintaining the responsible information appropriately [13]. In particular, the PFC had a causal role in the acquisition of new learning and fear conditioning, as well as the extinction of previously acquired learning [14,15]. The ventromedial PFC had a multifaceted functional role in regulating both fear extinction and conditioning [15,16], and the neuronal networks within the amygdala, the thalamus, the brainstem hippocampus, and the PFC were involved in the neural circuits of the acquisition of fear associative memory [17]. Perhaps, changes in the brain by Aβ, including the ACC and the PFC, could affect the specific functions of each region as consequential causes of cognitive impairment, a typical symptom of AD. In addition, approaches to the mechanism of cognitive impairment through fear-learning neural networks may contribute to the development of alternative and more accurate personalized therapies for AD as well as for psychiatric disorders [18].

Early-onset Alzheimer’s disease (EOAD) is a relatively rare neurogenetic cognitive disorder with a prevalence of <5%. Genetic variants in the *presenilin 1* (*PSEN1*), *presenilin 2* (*PSEN2*), and *APP* genes have been reported as the major causal factors of AD pathogenesis [19]. According to the Alzforum database (http://www.alzforum.org/mutations, accessed on 17 January 2022), there are more than 300 mutations in *PSEN1* and 76 mutations in *PSEN2*. The majority of EOAD cases have been identified in patients with a family history of dementia; however, several de novo cases have been identified. The presenilin 1 (PS1) and presenilin 2 (PS2) proteins were enzymatic components of γ-secretase, which could cleave and process the APP. Mutations in these genes could affect the production of Aβ peptide species from altered protein structures. In particular, *PSEN* mutations could contribute to AD pathogenesis by destabilizing the γ-secretase complex [20]. Pathogenic *PSEN* and *APP* mutations promoted the production and release of longer Aβ peptides by destabilizing the γ-secretase complex, thereby increasing AD risks. In AD, *PSEN2* mutations are rare, and few cases have been reported in Asia [21]. Carriers of *PSEN2* mutations tend to have a milder phenotype than patients with mutations in *PSEN1*. Furthermore, *PSEN2* mutations were associated with a broad age of disease onset, typically between 40 and 70 years [22,23].

This study aimed to describe clinical information of a Korean EOAD patient with a *PSEN2* Gly56Ser mutation. In addition, in silico analysis and structural change prediction were performed to address the pathogenicity of the mutation.

## 2. Results

### 2.1. Mutational Analysis

A heterozygous G > A substitution (chr1; rs188598190; c.G166A) was identified in *PSEN2* using Sanger sequencing (Figure 1A). The copresence of nucleotides was observed in both forward (G and A) and reverse (C and T) directions. The mutation caused a glycine (GGT, Gly, G) to serine (AGT, Ser, S) substitution within exon 5 of *PSEN2* (Figure 1B). Based on the NCBI Gene database, this mutation was predicted to occur at a very low frequency (0.000178, GnomAD; 0.000183, ExAC; 0.0007, KOREAN).

### 2.2. In Silico Assessment

The bulkiness score of mutated PS2 was greater than that of the wild-type protein (Figure 2A). In contrast, the hydrophobicity score (Eisenberg) of the mutant protein was markedly lower than that of the wild-type one (Figure 2C). No polarity score difference was observed when the mutant and wild-type proteins were compared (Figure 2B). Since a glycine residue was replaced with serine, it was predicted from PhosphoPICK analysis that phosphorylation by kinases, such as casein kinase II subunit alpha or casein kinase I isoform delta, may occur towards the cytosolic side. Phosphosite (https://www.phosphosite.org/homeAction, accessed on 3 December 2021) analysis predicted that most phosphorylation could occur within the N-terminal random coil region of proteins, where the N-terminal Ser56 would highly likely be phosphorylated.

From the 3D structural modeling of the secondary structure, Gly56 and Ser56 of the wild-type and mutant proteins, respectively, were predicted to be located within loop structures without definite structures (Figure 3A). Distinct properties of Gly and Ser could most likely alter the functions of the PS2 protein. Glycine is the smallest amino acid and serine is a polar amino acid with a hydroxymethyl group as its side chain, which could often form hydrogen bonds and phosphorylation. In the wild-type protein, Gly56 could interact exclusively with Ala45 (Figure 3B,C, left). Based on a model of the mutant protein, Ser56 presented the potential interactions with Met1, Ala6, Asp8, and Glu54 (Figure 3B,C, middle). Interestingly, in mutant model 3, the distance between the loop and the helix region in PS2 was shortened due to the formation of a hydrogen bond between Ser56 and Leu396 (Figure 3A–C, middle).

## 3. Discussion

Previously, our research group identified the following substitutions in PS2: Arg62Cys, His169Asn, and Val214Leu [21]. In addition, *PSEN2* mutations were reported in other studies. PS2 Arg62Cys was found in several patients with AD, which could change the populations of Aβ species in the cerebrospinal fluid (CSF) of patients [24,25,26]. However, this mutation was thought of as nonpathogenic, based on the presence in healthy controls with no changes in Aβ secretion, which was the same from a functional study with mouse neuroblastoma cells [27,28]. PS2 His169Asn and Val214Leu were located in a transmembrane region. Furthermore, His169Asn was found in several patients with AD, underscoring its high degree of pathogenicity [21]. Whether the newly identified Gly56Ser underlies AD pathophysiology remains uncertain. Although the patient had no family history of neurological disorders, we could not completely rule out the possibility of a hereditary link for AD, as the patient’s family did not consent to the release of information or genetic testing. According to the algorithm of Guerreiro, the Gly56Ser mutation was classified as possibly pathogenic [29].

In silico analysis revealed that the structural changes in the protein were likely caused by the Gly56Ser mutation. Since glycine is the smallest amino acid, the serine substitution with a hydroxymethyl side chain increased both the bulkiness and hydrophilicity (Figure 2). Three-dimensional structure prediction revealed that Ser56 was likely located within a random coil region with no secondary structure; therefore, it was not predicted to cause structural changes (Figure 3). However, Ser56 was predicted adequately to form a hydrogen bond with Leu396, consequently altering a closed N-terminal loop structure.

According to Alzforum, four cases were reported in which a presenilin protein mutation with a serine, threonine, or tyrosine substitution was capable of being phosphorylated in the cytosol-exposed N-terminal region (Table 1). These findings suggested that the pathology of AD could be mediated by presenilin protein phosphorylation due to amino acid substitution. A report that assessed cases of PS1 Asn24Ser from China in 2020 indicated that all the five carriers of the mutation among the eight offspring were diagnosed with AD [30]. All the patients had APOE ε3/3 type, and the average age of onset was 65.8 years. Moreover, although both non-carriers were relatively young, they were normal. Therefore, this mutation was determined to be potentially pathogenic [29]. PS1 Asn39Tyr was found in a patient with EOAD from England [31]. This mutation was classified as possibly deleterious according to the ACMG–AMP guidelines [32]. For PS2, only Gly34Ser mutation was reported from The Netherlands and China [27,30]. In particular, the Gly34Ser mutation was identified in three families in China. The average age at disease onset in these families was 61.6 years, and the presence of the mutation was confirmed in the affected family members by genotyping. PS2 Gly34Ser was not detected in any individual without AD among the family members. Interestingly, PS1 Arg41Ser was identified in patients with early-onset PD [33]. Pathogenic *PSEN* mutations were mostly found in transmembrane regions. However, previous reports suggested that N-terminal mutations (located in the cytosol) could have the potential to contribute to the pathogenesis of AD.

AD is an irreversible neurodegenerative disorder with no effective disease-modifying treatment. Constant efforts were poured to develop therapeutic agents worldwide. Aducanumab, which was recently approved conditionally by the FDA, is the antibody-based treatment by targeting Aβ oligomers [34]. Patients with mutations in representative genes, such as *APP*, *PSEN1*, and *PSEN2*, would have been affected in their Aβ production or aggregation, supporting the preferential treatment by Aβ-targeting therapeutics, including aducanumab. On the other hand, personalized treatment strategies would require various aspects due to the heterogeneity of AD. For instance, noninvasive brain stimulation (NIBS) techniques were reported for their potential for improving AD symptoms. NIBS improved the cognitive and memory functions and modulated fear memory in AD patients, and the utilization of biological markers and standardization of parameters were suggested to be used in large cohort studies [18,35]. This multifaceted treatment strategy could be an effective alternative towards personalized therapy according to the patient’s symptoms or the case of no response to drug treatment.

## 4. Materials and Methods

### 4.1. Patient Information

A 64-year-old right-handed woman with 12 years of education presented with a 4-year history of progressive cognitive impairment. Her symptoms occurred insidiously, with short-term memory loss and insomnia. The patient found difficulties in recalling events of the day. She regularly misplaced items and cooked the same foods repetitively on the same day. She also reported difficulty in recalling names and making simple calculations. However, her recalling of distant memories and visuospatial functioning were normal. She managed her household, bought groceries, and handled bank transactions alone without major problems. The patient and the caregivers did not report any personality changes or behavioral symptoms. She was diagnosed with hypertension and coronary heart disease and had no history of alcohol or drug abuse. No focal neurologic deficit was observed on neurological examination. Her father had no cognitive impairment up to his death at 91 years of age, and her mother died in her 70s of liver cancer. The patient had two sisters and two brothers (all aged 50–69), who were healthy and did not present any symptoms of cognitive impairment (Figure 4).

During the neuropsychological evaluation, the patient scored 24/30 in a Mini-Mental State Examination (MMSE). The subscores seven serial calculations were 3 out of 5 and the delayed word recall score was 0 out of 3, with a 0.5 Clinical Dementia Rating (CDR). The standardized neurological battery of the Seoul Neuropsychological Screening Battery (SNSB) was also performed. Her SNSB revealed that the digit span forward was 28.73%; the Korean Boston Naming Test result was 29.20%; Rey–Oesterrieth Complex Figure Test copy, 9.03%, calculation, <5%; Seoul Verbal Learning Test, delayed recall test, 5.17%, recognition test, 5.07%; Korean Stroop Test, color reading, 1.15%. She achieved normal results in spontaneous speech, reading, writing, and praxis tests. Neuropsychological tests revealed impairment in verbal memory, which was not improved by cues, and in executive functions. The routine laboratory findings were normal. Cerebrospinal fluid was not analyzed because the patient refused the test. Her apolipoprotein E genotype was 3/3. Brain magnetic resonance imaging showed that the volume of the bilateral hippocampus was reduced, without any ischemic changes or lesions (Figure 5A). 18F-Fluorodeoxyglucose positron emission tomography (FDG-PET) showed bilateral temporoparietal association cortices, precuneus, inferior parietal lobule, and middle temporal gyrus hypometabolism, a typical pattern of glucose metabolism in patients with AD (Figure 5B). The patient was initially diagnosed with probable AD using the National Institute on Aging–Alzheimer’s Association (NIA–AA) criteria. She was prescribed donepezil, an anti-acetylcholinesterase inhibitor. For accurate disease diagnosis, extensive genetic screening was performed using whole-exome sequencing analysis. The family members refused genetic testing.

### 4.2. DNA Purification and Genetic Screening

A blood sample (5 mL) was collected from the patient in an EDTA-containing tube and stored at −20 °C for further use. Genomic DNA from blood cells was purified using a QIAamp DNA Blood Maxi Kit (Qiagen, Hilden, Germany) according to the manufacturer’s instructions. Purified DNA was stored at −20 °C until analysis. Whole-exome sequencing was performed to screen for variants in the genes associated with AD. Purified DNA samples were quantified and used for library construction. The Illumina platform was applied for sequencing before bioinformatics analysis. The fluorescence files on the Illumina platform were transformed by base calling into short reads recorded in the FASTQ format containing sequence information. Raw sequencing data were aligned to the human reference genome (hg19) to detect variants in the sample. PCR-based genetic analysis was conducted to verify the presence of the mutation. Standard Sanger sequencing was carried out in both directions by Bioneer Inc. (Daejeon, Korea). BigDye Terminator cyclic sequencing was performed using an ABI 3730XL DNA Analyzer (Bioneer Inc., Daejeon, Korea). Sequences of the genes and proteins were checked using the NCBI Gene (http://www.ncbi.nlm.nih.gov/gene, accessed on 15 October 2021) and UniProt (http://www.uniprot.org, accessed on 15 October 2021) databases, respectively. 

### 4.3. In Silico Analyses

Changes due to a mutation in the PS2 protein were analyzed to evaluate the pathogenicity of the mutation. Changes in the bulkiness, polarity, and hydrophobicity of PS2 were analyzed using ExPAsy (https://www.expasy.org/, accessed on 2 November 2021) tools. Since the PS2 mutation Gly56Ser was determined to be exposed to the cytosol, phosphorylation by kinases was predicted using PhosphoPICK [36]. The predicted 3D structures of the wild-type and mutated PS2 proteins were assessed using the Raptor X software (http://raptorx.uchicago.edu/, accessed on 21 November 2021). The predicted interactions with the surrounding amino acids were assessed using the Discovery Studio 3.5 Visualizer software. The overall process of this study was schematically shown in Figure 6.

## 5. Conclusions

This study indicated that a *PSEN2* Gly34Ser mutation contributed to AD pathogenicity in EOAD patients who presented with typical AD symptoms and brain degeneration. Substitution with serine in a position facing the cytosol had the potential to affect the function of PS2 through interactions with other residues and/or phosphorylation. Previous studies reported that substitutions involving amino acids capable of being phosphorylated at the N-terminus of the presenilin proteins presented tight familial links; therefore, *PSEN2* Gly34Ser may promote AD pathogenicity with similar mechanisms. Future functional studies are needed to fully determine the disease progression and mechanisms.

## 6. Limitations and Future Directions

Segregation analysis of the PS2 Gly34Ser mutation could not be performed because all the living family members refused genetic testing. The relationship between disease phenotypes and mutations was validated through additional clinical and genetic studies. Immunohistochemistry and the measurements of Aβ or tau were not performed, since no brain or cerebrospinal fluid samples from the patient were available. Transfected cells with PS2 Gly34Ser mutations could help to investigate the underlying mechanisms of influence on AD pathogenesis in the future.

## Figures and Tables

**Figure 1 ijms-23-02967-f001:**
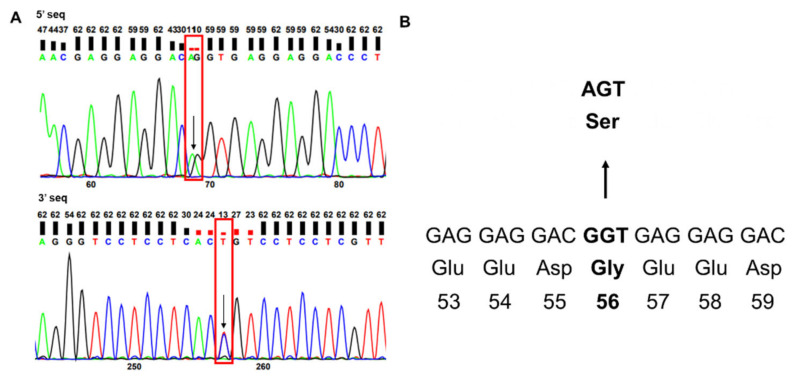
*PSEN2* sequencing data revealing a mutation resulting in a Gly56Ser substitution are shown. (**A**) Sanger sequencing data and the (**B**) location of the Gly56Ser substitution in *PSEN2*. Arrows indicate the mutation positions and the altered sequence and amino acid by mutation.

**Figure 2 ijms-23-02967-f002:**
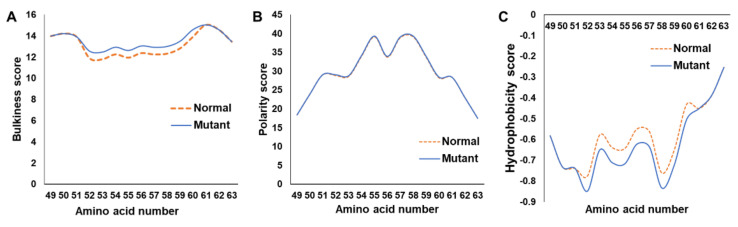
ExPASY scores of wild-type and Gly56Ser PS2. The comparison of protein bulkiness (**A**), polarity (**B**), and hydrophobicity (**C**) scores of wild-type and mutant PS2 via the ExPASY analysis.

**Figure 3 ijms-23-02967-f003:**
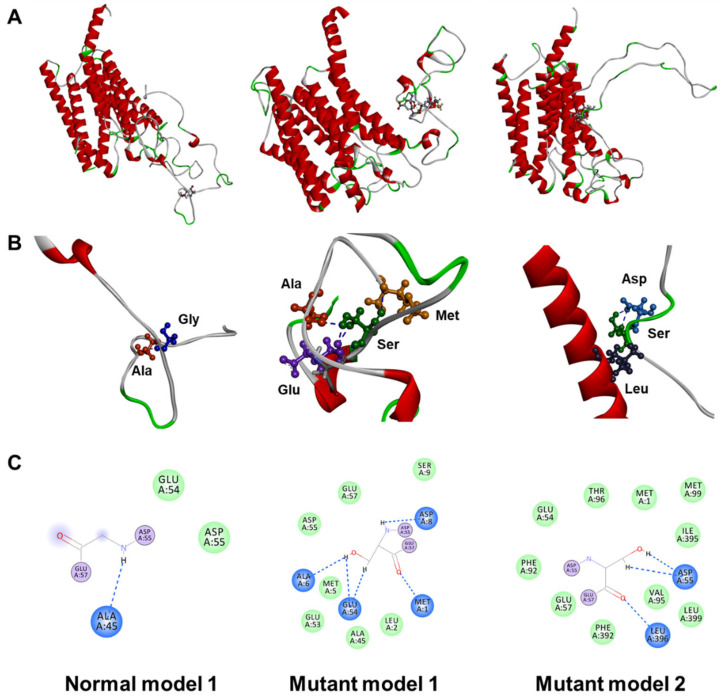
Three-dimensional predicted structure and possible intramolecular interactions of PS2 Gly56Ser. (**A**) Predicted 3D structures of the wild-type and mutant PS2 proteins. (**B**) Enlarged images of specific interaction areas and (**C**) 2D diagram of amino acid interactions. Blue indicates hydrogen bonds and green represents van der Waals forces.

**Figure 4 ijms-23-02967-f004:**
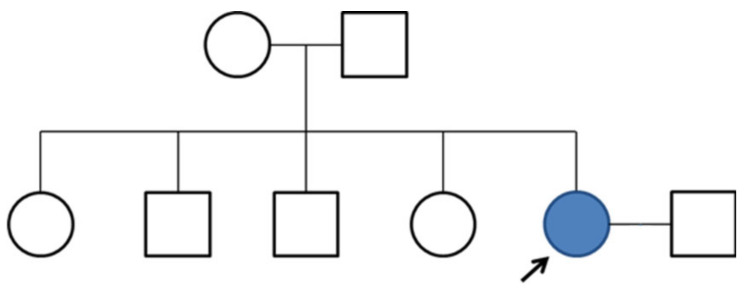
A family pedigree of the patient with Alzheimer’s disease, and a PS2 Gly56Ser mutation carrier is indicated.

**Figure 5 ijms-23-02967-f005:**
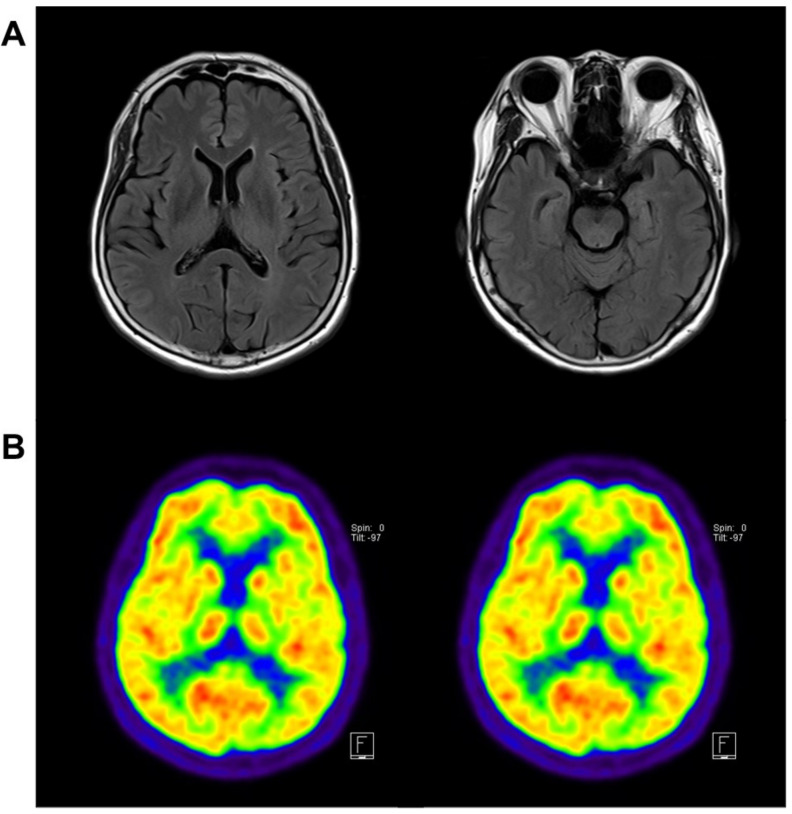
Imaging data of the patient with a PS2 Gly56Ser mutation. (**A**) MRI findings revealing mild parietal and hippocampal atrophy and (**B**) FDG-PET findings showing temporoparietal hypometabolism. Abbreviations: MRI, magnetic resonance imaging; FDG-PET, 18F-fluorodeoxyglucose positron emission tomography.

**Figure 6 ijms-23-02967-f006:**
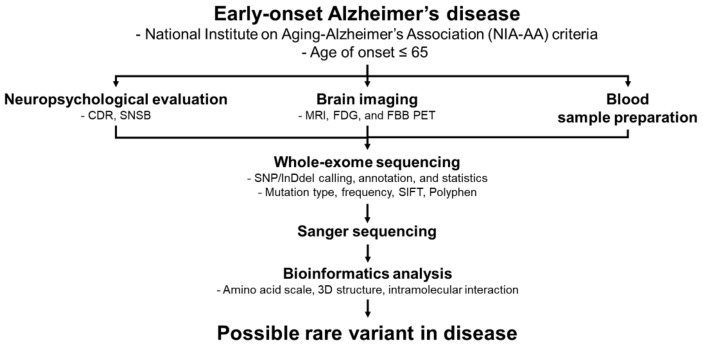
The flow chart describes the major experimental steps from the AD patient to the identification of mutation.

**Table 1 ijms-23-02967-t001:** Mutations substituted with phosphorylation-capable residues in the cytosolic N-terminal region of presenilin.

Gene	Mutation	Age of Onset (y)	Family History	Functional Data	Clinical Phenotype
*PSEN1*	Asn24Ser	65.8	Familial (five carriers with AD from China)	Damaging effect predicted by at least two algorithms	AD
	Asn39Tyr	Not available	Not available	Conflicting results from in silico predictive analyses	EOAD
	Arg41Ser	35	No family history	Cerebrospinal fluid levels of Aβ and tau were normal; phospho-tau levels were elevated	EOPD
*PSEN2*	Gly34Ser	61.6 (China)	Familial (three families and ten carriers with AD or MCI from China	Unchanged Aβ42/Aβ40 ratio (The Netherlands)	LOAD (The Netherlands) EOAD and LOAD (China)

## Data Availability

The data presented in this study are available on request from the corresponding author.

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
