# Peer review of "A Possible Pathogenic PSEN2 Gly56Ser Mutation in a Korean Patient with Early-Onset Alzheimer’s Disease"

_ijms, 2022, doi:10.3390/ijms23062967_

Round 1

Reviewer 1 Report

This paper is technically sound and interesting.  This paper is fine as a case report of PSEN2 Gly56Ser mutation as judged by MRI and PET, but the authors did not mention any  clinical symptoms by which  mild cognitive impairment or AD was suspected.

In conclusion before publication  the authors should mention at least some clinical symptom of this patient. In this patient no family history was available and  It is difficult for readers to understand the significance of this mutation.  Probably no specific symptoms were found but if so at least this should be described. 

Author Response

Response to Reviewer 1 Comments

Point 1: This paper is technically sound and interesting. This paper is fine as a case report of PSEN2 Gly56Ser mutation as judged by MRI and PET, but the authors did not mention any clinical symptoms by which mild cognitive impairment or AD was suspected. In conclusion before publication the authors should mention at least some clinical symptom of this patient. In this patient no family history was available and  It is difficult for readers to understand the significance of this mutation. Probably no specific symptoms were found but if so at least this should be described.

Response 1: We appreciate your valuable suggestion. Additional symptoms and clinical parameters were mentioned to support the clinical diagnosis and the significance of the mutation (p. #6, lines #208-#214; p. #7, lines #225-#234).

‘The patient found difficulties in recalling events of the day. She regularly misplaced items and cooked the same foods repetitively on the same day. She also reported dif-ficulty in recalling name and making simple calculations. However, her recall of distant memories and visuospatial functioning were normal. She managed her household, bought groceries and handled the bank transaction alone without major problems. The patient and caregivers did not report any personality changes or behavioral symptoms.’

‘From the neuropsychological evaluation, the patient scored 24/30 on a Mini-Mental State Exam-ination (MMSE). The subcored seven serial calculations were 3 out of 5 and delayed word recall scores was 0 out of 3 with an 0.5 Clinical Dementia Rating (CDR). The standardized neurological battery of the Seoul Neuropsychological Screening Battery (SNSB) was also performed. Her SNSB revealed that the digit span forward was 28.73%; Korean-Bost Naming Test was 29.20%; Rey-Oesterrieth Complex Figure Test copy, 9.03%, calculation, <5%, Seoul Verbal Learning Test-delayed recall test, 5.17%, and recognition test was 5.07%; and Korean Stroop-test-color reading, 1.15%. She showed normal in spontaneous speech, reading, writing, and praxis tests. Neuropsychological tests revealed impairment in verbal memory, which was not improved by cues, and in executive functions.’

Reviewer 2 Report

This research article by Shim and colleagues, entitled ‘A possible pathogenic PSEN2 Gly56Ser mutation in a Korean patient with early-onset Alzheimer’s disease’, does an excellent work demonstrating the presence of a PSEN2 point mutation (resulting in a nucleotide change from G to A at position 166) in a 64-year-old Korean female with Alzheimer's disease (AD). DNA purification, genetic screenings, and positron emission tomography tests were conducted: results showed hippocampal atrophy, temporal and parietal hypometabolism, and how the mutation altered protein bulkiness and hydrophobicity of the mutant region of the protein.

The main strength of this functional study is that it addresses an interesting and trending topic, investigating rare genetic variants of unknown clinical significance in AD: authors described clinical information of a Korean patient with early-onset Alzheimer’s disease who had a Gly56Ser substitution in PSEN2, and explored the pathogenicity of the mutation. In general, I think the idea of this article is really interesting and the authors’ fascinating observations on this timely topic may be of interest to the readers of International Journal of Molecular Sciences. However, some comments, as well as some crucial evidence that should be included to support the authors’ argumentation, needed to be addressed to improve the quality of the article, its adequacy, its replicability, and thus its readability prior to the publication in the present form. My overall judgment is to publish this article after the authors have carefully considered my suggestions below, in particular reshaping parts of the Introduction and Discussion sections, and by adding more evidence.

Please consider the following comments:

  • Keywords: Please consider adding ‘hippocampal atrophy and ‘PSEN2 Gly56Ser mutation’ as keywords.
  • The Manuscript: I recommend authors to include more evidence to back their claims, especially in the Introduction of the article, which I believe is seriously lacking. Thus, I recommend the authors to focus on deepening the subject of their manuscript, as the bibliography is too concise: nonetheless, in my opinion, less than 50 articles for a research paper are really insufficient. Currently, authors cite only 19 papers, and they are dramatically few. Therefore, I suggest the authors to focus their efforts on researching relevant literature: I believe that adding more studies will help to provide better and more accurate background to this study. In this review, I will try to help the authors by suggesting relevant literature that suits their manuscript.
  • Introduction: The ‘Introduction’ section is well-written and nicely presented, with a good balance of descriptive text about molecular and cell biological aspects of AD, and interpretative illustration of Aβ peptide production’s role in the pathogenesis of Alzheimer's disease. Nevertheless, I think that more information about pathophysiology and neurological changes of Alzheimer’s disease would provide a better background here. Thus, I suggest the authors to make an effort to provide a brief overview of the pertinent published literature that offers a perspective on structural and functional correlates of age-associated cognitive changes that might indicate neurodegeneration and lead to dementia because as it stands, this information is not highlighted in the text. In this regard, I believe that the statement ‘Aβ peptides, the most important causative factor for AD, induces neurotoxicity by forming oligomers via self-aggregation’ needs some necessary citations. In particular, according with this sentence, I would recommend citing a recent review that examined pathophysiological basis and biomarkers of AD pathology and investigated molecular signs of neuroinflammation in neurodegenerative diseases, in particular Alzheimer’s disease (https://doi.org/10.3390/ijms21072431). I also recommend a relevant study in which authors investigated age-related impairments in the ability to process contextual information and in the regulation of responses to threat, addressing that structural and physiological alterations in the prefrontal cortex and medial temporal lobe determine cognitive changes in advanced aging, that can eventually cause patterns of cognitive dysfunctions observed in patients with AD/MCI (https://doi.org/10.1038/s41598-018-31000-9). I firmly believe that these improvements will help to provide a more coherent and defined background.
  • Introduction: In according with the previous point raised, when authors stated that ‘…changes in the Aβ peptide and γ-secretase complex alter aggregation properties and Aβ production, resulting in increased amount of Aβ oligomer forms. Therefore, mutations in amyloid precursor protein (APP) and components of the γ-secretase complex may facilitate AD by increasing levels of Aβ peptide aggregation and production’, I would suggest adding some evidence that might address how forms of Aβ and tau protein work together to drive healthy neurons into the diseased state, consistently in the frontal and/or parietal lobes, causing alterations of frontal lobe that impact memory and error-driven learning in individuals who have a high risk of dementia, may improve the theoretical background of the present article and its argumentation: evidence from an electrophysiological study suggested that medio-frontal ERP signals of prediction error tracks the timing of salient events, and highlighted how alterations in medial prefrontal cortex could impact on the patients’ capacity of signal errors in the prediction of outcomes (https://doi.org/10.1162/jocn_a_01074). Importantly, evidence from a recent study conducted on patients with lesion in ventromedial portion of prefrontal cortex (https://doi.org/10.1523/JNEUROSCI.0304-20.2020) revealed that the ventromedial prefrontal cortex (vmPFC) is involved in the acquisition of emotional conditioning (i.e., learning), assessing how naturally occurring bilateral lesion centered on the vmPFC compromises the generation of a conditioned psychophysiological response during the acquisition of pavlovian threat conditioning (i.e., emotional learning). Finally, in a recent theoretical review that focused on the neurobiology of conditioning (i.e., learning), the role of the ventromedial prefrontal cortex (vmPFC) was analyzed in the processing of safety-threat information and their relative value, and how this region is fundamental for the evaluation and representation of stimulus-outcome’s value needed to produce sustained physiological responses (https://doi.org/10.1038/s41380-021-01326-4). Finally, authors might also see studies that have focused on this topic (https://doi.org/10.1162/NECO_a_00779; https://doi.org/10.1038/s41386-021-01101-7).
  • Results: Please provide more statistical details to ensure in-depth understanding and replicability of the findings. Specifically, provide more detail about mutational analysis, because it appears unclear and hard to grasp how to interpret variants at tissue and single-cell resolution.
  • Discussion: In this section, the authors thoroughly described the results and their argumentation and captured the state of the art well; however, I would have liked to see some views on a way forward. Hence, I ask them to include some thought as well as in-depth considerations, making an effort, trying to explain the theoretical as well as the translational application of their research.
  • Also, even though it is not mandatory, I believe that a ‘Conclusion’ section would be useful to adequately convey what the author believe is the take-home message of their study, and therefore provide a synthesis of the data presented in the paper.
  • In according to the previous comment, I believe that the authors should make an effort, trying to explain the theoretical implication as well as the translational application of this case study, to adequately convey what they believe is the take-home message of their study, and therefore discussing theoretical and methodological avenues in need of refinement, suggesting a path forward in the understanding of cellular senescence in neurons and glial cells’ putative role in AD. In this regard, recent evidence suggests that the application of new methods in Alzheimer’s treatment, such as the Non-invasive brain stimulation techniques (NIBS), have shown promising results in humans (https://doi.org/1097/WCO.0000000000000669). Importantly, I recommend referring to recent studies that revealed that the application of NIBS induces long-lasting effects, noninvasively modulating the cortical excitability, and modulating a variety of cognitive functions: for example, a recent review acknowledged the implementation of NIBS to modulate in general fear memories (https://doi.org/10.1016/j.neubiorev.2021.04.036). Additionally, I would suggest another recent study that illustrated the therapeutic potential of NIBS as a valid alternative for those patients not responding or drug treatments (https://doi.org/10.1016/j.jad.2021.02.076).
  • In according to the previous comment, I would ask the authors to include a proper ‘Limitations and future directions’ section before the end of the manuscript, in which authors can describe in detail and report all the technical issues brought to the surface.
  • In Silico analyses: I would suggest the authors to try employ computational protein design techniques that can allow for the in silico evaluation of PS2 protein mutation, to check the probable causality of mutations encountered in patients at the protein and at the DNA levels.
  • Figures: I suggest to modify Figures 1 – 2 – 3 for clarity and provide higher-quality images because, as it stands, the readers may have difficulty comprehending them. In my opinion, data settings are written with a very small font. Also, please change the scale of the vertical axis and use the same minimum/maximum scale value in all the graphs in all the figures and reorganize the graphs’ space, to provide a better understanding and a direct interpretation of the results. Finally, I suggest adding a figure that can clearly describes experimental design.
  • References: According to the Journal’s guidelines, authors should have provided the DOI number for each reference.

Overall, the manuscript contains 5 figures, 1 table, and 19 references. In my opinion, the number of references is too low for an original research article, and this issue may prevent the possibility of publishing it in this form. However, I believe that the manuscript may carry important value in presenting clinical information of a Korean patient with EOAD 59 who had a Gly56Ser substitution in PSEN2.

I hope that, after these careful revisions, the manuscript can meet the Journal’s high standards for publication. I am available for a new round of revision of this manuscript.

Best regards,

Reviewer

Author Response

Response to Reviewer 2 Comments

Point 1: Keywords: Please consider adding ‘hippocampal atrophy and ‘PSEN2 Gly56Ser mutation’ as keywords.

Response 1: Thanks for your comments. We added two suggested keywords in the manuscript.

Point 2: The Manuscript: I recommend authors include more evidence to back their claims, especially in the Introduction of the article, which I believe is seriously lacking. Thus, I recommend the authors to focus on deepening the subject of their manuscript, as the bibliography is too concise: nonetheless, in my opinion, less than 50 articles for a research paper are really insufficient. Currently, authors cite only 19 papers, and they are dramatically few. Therefore, I suggest the authors to focus their efforts on researching relevant literature: I believe that adding more studies will help to provide better and more accurate background to this study. In this review, I will try to help the authors by suggesting relevant literature that suits their manuscript.

Response 2: Thank you for raising the important points. As following the reviewer’s comments, we supplemented more background with the relevant literature in the introduction section.

Point 3: Introduction: The ‘Introduction’ section is well-written and nicely presented, with a good balance of descriptive text about molecular and cell biological aspects of AD, and interpretative illustration of Aβ peptide production’s role in the pathogenesis of Alzheimer's disease. Nevertheless, I think that more information about pathophysiology and neurological changes of Alzheimer’s disease would provide a better background here. Thus, I suggest the authors to make an effort to provide a brief overview of the pertinent published literature that offers a perspective on structural and functional correlates of age-associated cognitive changes that might indicate neurodegeneration and lead to dementia because as it stands, this information is not highlighted in the text. In this regard, I believe that the statement ‘Aβ peptides, the most important causative factor for AD, induces neurotoxicity by forming oligomers via self-aggregation’ needs some necessary citations. In particular, according with this sentence, I would recommend citing a recent review that examined pathophysiological basis and biomarkers of AD pathology and investigated molecular signs of neuroinflammation in neurodegenerative diseases, in particular Alzheimer’s disease (https://doi.org/10.3390/ijms21072431). I also recommend a relevant study in which authors investigated age-related impairments in the ability to process contextual information and in the regulation of responses to threat, addressing that structural and physiological alterations in the prefrontal cortex and medial temporal lobe determine cognitive changes in advanced aging, that can eventually cause patterns of cognitive dysfunctions observed in patients with AD/MCI (https://doi.org/10.1038/s41598-018-31000-9). I firmly believe that these improvements will help to provide a more coherent and defined background.

Introduction: In according with the previous point raised, when authors stated that ‘…changes in the Aβ peptide and γ-secretase complex alter aggregation properties and Aβ production, resulting in increased amount of Aβ oligomer forms. Therefore, mutations in amyloid precursor protein (APP) and components of the γ-secretase complex may facilitate AD by increasing levels of Aβ peptide aggregation and production’, I would suggest adding some evidence that might address how forms of Aβ and tau protein work together to drive healthy neurons into the diseased state, consistently in the frontal and/or parietal lobes, causing alterations of frontal lobe that impact memory and error-driven learning in individuals who have a high risk of dementia, may improve the theoretical background of the present article and its argumentation: evidence from an electrophysiological study suggested that medio-frontal ERP signals of prediction error tracks the timing of salient events, and highlighted how alterations in medial prefrontal cortex could impact on the patients’ capacity of signal errors in the prediction of outcomes (https://doi.org/10.1162/jocn_a_01074). Importantly, evidence from a recent study conducted on patients with lesion in ventromedial portion of prefrontal cortex (https://doi.org/10.1523/JNEUROSCI.0304-20.2020) revealed that the ventromedial prefrontal cortex (vmPFC) is involved in the acquisition of emotional conditioning (i.e., learning), assessing how naturally occurring bilateral lesion centered on the vmPFC compromises the generation of a conditioned psychophysiological response during the acquisition of pavlovian threat conditioning (i.e., emotional learning). Finally, in a recent theoretical review that focused on the neurobiology of conditioning (i.e., learning), the role of the ventromedial prefrontal cortex (vmPFC) was analyzed in the processing of safety-threat information and their relative value, and how this region is fundamental for the evaluation and representation of stimulus-outcome’s value needed to produce sustained physiological responses (https://doi.org/10.1038/s41380-021-01326-4). Finally, authors might also see studies that have focused on this topic (https://doi.org/10.1162/NECO_a_00779; https://doi.org/10.1038/s41386-021-01101-7).

Response 3: As a reviewer’s suggestion, we have added the content in the proper position. We described possible downstream pathways driven from Aβ species that result in the structural and physiological changes in the brain (p. #1, lines #39-#46). Also, we stated the impact of memory and error-driven learning in individuals induced by abnormal Aβ accumulation in the prefrontal cortex and other relevant brain regions (p. #2, lines #51-#62). Because the anterior cingulate cortex and the prefrontal cortex are key components of cognitive networks, Aβ accumulation in those regions may cause cognitive impairment, a typical symptom of AD.

‘Aβ also may trigger the microglia activation and reactive astrocyte proliferation, leading to pro-inflammatory cytokines releases, such as interleukin (IL)-1β and tumor necrosis factor (TNF)-α, inducing neuroinflammation [3]. In addition, Aβ could cause neuronal cell death and cognitive impairment through several mechanisms, including glutami-nase down-regulation and Ca2+ flux induction [4]. A recent study addressed that struc-tural and physiological changes in the frontal and medial temporal lobes under the ef-fect of normal aging determined cognitive alternation in aging [5], which could be ex-acerbated by intrinsic or extrinsic properties of Aβ.’

‘Aβ would be deposited mainly in brain regions following, the hippocampus, amygdala, and entorhinal cortex [6]. Abnormal cognitive processes in AD patients could be cor-related with Aβ accumulations in the prefrontal cortex (PFC) and other relevant brain regions [7]. The anterior cingulate cortex (ACC) and PFC were consistently associated with the salience detection, error prediction, error monitoring, monitoring of undesir-able outcomes, and major depressive disorder [8-11]. ACC and PFC were key compo-nents of cognitive networks, and were activated routinely during monitoring of the external environments and maintaining responsible information appropriately [12]. In particular, PFC had a causal role in the acquisition of new learning and fear conditioning, as well as, the extinction of previously acquired learning [13, 14]. Perhaps, changes in the brain by Aβ, including ACC and PFC, could affect the specific functions of each region, as consequential causes in cognitive impairment, a typical symptom of AD.’

Point 4: Results: Please provide more statistical details to ensure in-depth understanding and replicability of the findings. Specifically, provide more detail about mutational analysis, because it appears unclear and hard to grasp how to interpret variants at tissue and single-cell resolution.

Response 4: Thank you for your opinion. Variants were investigated by whole exome sequencing technics with the DNA purified from the patient’s blood cells. The process of whole exome sequencing was added in the method part to provide more detail about mutational analysis (p. #8, lines #256-#262).

‘Whole-exome sequencing was performed to screen for variants in genes associated with AD. Purified DNA samples were quantified and used for library construction. The Il-lumina platform was applied for sequencing before bioinformatics analysis. The fluo-rescence files on the Illumina platform were transformed by base calling into short reads recorded in FASTQ format containing sequence information. Raw sequencing data were aligned to the human reference genome (hg19) to detect variants in the sample.’

Point 5: Discussion: In this section, the authors thoroughly described the results and their argumentation and captured the state of the art well; however, I would have liked to see some views on a way forward. Hence, I ask them to include some thought as well as in-depth considerations, making an effort, trying to explain the theoretical as well as the translational application of their research.

Also, even though it is not mandatory, I believe that a ‘Conclusion’ section would be useful to adequately convey what the author believe is the take-home message of their study, and therefore provide a synthesis of the data presented in the paper.

In according to the previous comment, I believe that the authors should make an effort, trying to explain the theoretical implication as well as the translational application of this case study, to adequately convey what they believe is the take-home message of their study, and therefore discussing theoretical and methodological avenues in need of refinement, suggesting a path forward in the understanding of cellular senescence in neurons and glial cells’ putative role in AD. In this regard, recent evidence suggests that the application of new methods in Alzheimer’s treatment, such as the Non-invasive brain stimulation techniques (NIBS), have shown promising results in humans (https://doi.org/1097/WCO.0000000000000669). Importantly, I recommend referring to recent studies that revealed that the application of NIBS induces long-lasting effects, noninvasively modulating the cortical excitability, and modulating a variety of cognitive functions: for example, a recent review acknowledged the implementation of NIBS to modulate in general fear memories (https://doi.org/10.1016/j.neubiorev.2021.04.036). Additionally, I would suggest another recent study that illustrated the therapeutic potential of NIBS as a valid alternative for those patients not responding or drug treatments (https://doi.org/10.1016/j.jad.2021.02.076).

In according to the previous comment, I would ask the authors to include a proper ‘Limitations and future directions’ section before the end of the manuscript, in which authors can describe in detail and report all the technical issues brought to the surface.

Response 5: Thank you for your valuable comments. As suggested by the reviewer, we added the ‘conclusion’ and ‘Limitations and future directions’ sections. Aducanumab, an FDA-approved drug targeting Aβ was mentioned, and we suggested that treatment strategies in various directions such as NIBS are needed for personalized therapy due to the heterogeneity of AD (p. #5, lines #169-#182). We also have stated the limitations of this study and described future plans to address these issues (p. #6, lines #196-#203).

‘AD is an irreversible neurodegenerative disorder with no effective disease modi-fying treatment. Constant efforts were poured to develop therapeutic agents worldwide. Aducanumab, which was recently approved conditionally by the FDA, was the anti-body-based treatment by targeting Aβ oligomers [30]. Patients with mutations in rep-resentative genes, such as APP, PSEN1, and PSEN2, would have been affected in their Aβ productions or aggregations, supporting the preferential treatment by Aβ targeting therapeutics, including Aducanumab. On the other hand, the personalized treatment strategies would require various aspects due to the heterogeneity of AD. For instance, noninvasive brain stimulation (NIBS) techniques were reported for their potential for improving AD symptoms. NIBS improved the cognitive and memory functions and modulated fear memory in AD patients, and the utilization of biological markers and standardization of parameters were suggested to be used in large cohort studies [31, 32]. This multi-faceted treatment strategy could be an effective alternative towards person-alized therapy according to the patient's symptoms or the case of no response to drug treatment [33].’

‘Segregation analysis of PS2 Gly34Ser mutation could not be performed because all living family members refused the genetic testing. The relationship between disease phenotypes and mutations was validated through additional clinical and genetic studies. Immunohistochemistry and the measurements of Aβ or tau were not performed, since no brain or cerebrospinal fluid samples from the patient were available. Transfected cells with PS2 Gly34Ser mutations could help to investigate the underlying mechanisms of influence on AD pathogenesis in the future.’

Point 6: In Silico analyses: I would suggest the authors to try employ computational protein design techniques that can allow for the in silico evaluation of PS2 protein mutation, to check the probable causality of mutations encountered in patients at the protein and at the DNA levels.

Response 6: Thank you for your opinion. We previously performed in silico analysis to predict the probable causality of mutation in PS2 protein. Changes in bulkiness, polarity, and hydrophobicity were compared between wild-type and mutated PS2 proteins using ExPAsy tools. The polarity was not changed, but hydrophobicity and bulkiness were altered by the substitution with a serine residue instead of glycine (Figure 2). In addition, phosphorylation on the substituted serine residue by various kinases was anticipated by PhosphoPICK. Possible changes of protein 3D structure and intramolecular interactions were predicted and visualized using Raptor X software (computational protein design techniques) and Discovery Studio 3.5 Visualizer software (Figure 3). We hope that the conducted in silico analysis would be sufficient to support the possible pathogenicity of the mutation.

Point 7: Figures: I suggest to modify Figures 1 – 2 – 3 for clarity and provide higher-quality images because, as it stands, the readers may have difficulty comprehending them. In my opinion, data settings are written with a very small font. Also, please change the scale of the vertical axis and use the same minimum/maximum scale value in all the graphs in all the figures and reorganize the graphs’ space, to provide a better understanding and a direct interpretation of the results. Finally, I suggest adding a figure that can clearly describes experimental design.

Response 7: Thank you for your opinion. Although we increased the font size in Figure 2, Figures 1 and 3 inevitably could not increase the size because these figures are a non-modifiable format. Each graph in Figure 2 represented different results, so it would be proper for using different minimum/maximum scale values as previously published papers [1-4]. As the reviewer’s suggestion, we made a flow chart to help readers understand our experimental design (Figure 6).

  1. Bagyinszky, E.; Ch’ng, G.-S.; Chan, M.-Y.; An, S.S.A.; Kim, S. A Pathogenic Presenilin-1 Val96Phe Mutation from a Malaysian Family. Brain Sci. 2021, 11, 1328. https://doi.org/10.3390/brainsci11101328
  2. Senanarong, V.; An, S.S.A.; Giau, V.V.; Limwongse, C.; Bagyinszky, E.; Kim, S. Pathogenic PSEN1 Glu184Gly Mutation in a Family from Thailand with Probable Autosomal Dominant Early Onset Alzheimer’s Disease. Diagnostics 2020, 10, 135. https://doi.org/10.3390/diagnostics10030135
  3. Bagyinszky, E.; Lee, H.; Pyun, J.M.; Suh, J.; Kang, M.J.; Vo, V.G.; An, S.S.A.; Park, K.H.; Kim, S. Pathogenic PSEN1 Thr119Ile Mutation in Two Korean Patients with Early-Onset Alzheimer’s Disease. Diagnostics 2020, 10, 405. https://doi.org/10.3390/diagnostics10060405
  4. Bagyinszky E, Yang Y, Giau VV, Youn YC, An SSA, Kim S. Novel prion mutation (p.Tyr225Cys) in a Korean patient with atypical Creutzfeldt–Jakob disease. Clin Interv Aging. 2019;14:1387-1397

https://doi.org/10.2147/CIA.S210909

Figure 6 The flow chart describes the major experimental steps from AD patient to the identification of mutation.

Point 8: References: According to the Journal’s guidelines, authors should have provided the DOI number for each reference.

Response 8: Based on the Journal’s guidelines, we have added the DOI number for each reference.

Round 2

Reviewer 2 Report

I am very pleased to see that the authors have welcomed many of my suggestions and have clarified several of the questions I raised in my first round of this review. I believe that this functional study does an excellent work investigating rare genetic variants of unknown clinical significance in AD, describing early-onset Alzheimer’s disease in a patient with Gly56Ser substitution in PSEN2.

I only have two last minor suggestions to assess before endorsing publishing, to further improve the theoretical background of the present article and its argumentation by highlighting how cognitive alterations caused by frontal dysfunction are fundamental as neurodegenerative biomarkers of AD. In this regard, I suggest adding evidence from a recent theoretical review that focused on the neurobiology of conditioning (i.e., learning), analyzing the role of the ventromedial prefrontal cortex (vmPFC) in the processing of safety-threat information and their relative value (https://doi.org/10.1038/s41380-021-01326-4). Furthermore, I would suggest adding information from a very recent perspective manuscript (https://doi.org/10.17219/acem/146756) that has focused on providing a deeper understanding of human learning neural networks, showed the crucial role of human PFC, giving interesting insights on the involvement of this important brain region in the advancement of alternative, more precise and individualized treatments for a variety of neurologic and psychiatric disorders.

Overall, this is a timely and needed study, and I look forward to seeing further study on this issue by these authors in the future.

Author Response

Response to Reviewer 2 Comments

Point 1: I am very pleased to see that the authors have welcomed many of my suggestions and have clarified several of the questions I raised in my first round of this review. I believe that this functional study does an excellent work investigating rare genetic variants of unknown clinical significance in AD, describing early-onset Alzheimer’s disease in a patient with Gly56Ser substitution in PSEN2.

I only have two last minor suggestions to assess before endorsing publishing, to further improve the theoretical background of the present article and its argumentation by highlighting how cognitive alterations caused by frontal dysfunction are fundamental as neurodegenerative biomarkers of AD. In this regard, I suggest adding evidence from a recent theoretical review that focused on the neurobiology of conditioning (i.e., learning), analyzing the role of the ventromedial prefrontal cortex (vmPFC) in the processing of safety-threat information and their relative value (https://doi.org/10.1038/s41380-021-01326-4). Furthermore, I would suggest adding information from a very recent perspective manuscript (https://doi.org/10.17219/acem/146756) that has focused on providing a deeper understanding of human learning neural networks, showed the crucial role of human PFC, giving interesting insights on the involvement of this important brain region in the advancement of alternative, more precise and individualized treatments for a variety of neurologic and psychiatric disorders.

Response 1: Thanks for your efforts in improving the theoretical background of our manuscript. As the reviewer’s suggestion, we additionally have added the neurobiology of conditioning related to PEC (p. #2, lines #62-#66). We also have stated more precise and personalized treatments could be applied by focusing on the fear learning neural networks (p. #2, lines #68-#71).

‘The ventromedial PFC had a multifaceted functional role in regulating both fear extinction and conditioning [16, 17], and the neuronal networks within amygdala, thalamus, brain-stem hippocampus, and PFC were involved in the neural circuits of the acquisition of fear associative memory [18].’

‘In addition, approaches to the mechanism of cognitive impairment through fear learning neural networks may contribute to the development of alternative and more accurate personalized therapies for AD as well as psychiatric disorders [19].’

Point 2: Overall, this is a timely and needed study, and I look forward to seeing further study on this issue by these authors in the future.

Response 2: We have routinely been functional studies by applying the mutation sequence in the human cell lines. Other mutations are on progress and will be published soon. PS2 G56S mutation was also arranged in our list of possible pathogenic variants. We also hope and are looking forward to proving the pathogenicity of this mutation by our functional study in the future.

This manuscript is a resubmission of an earlier submission. The following is a list of the peer review reports and author responses from that submission.